# Long-Term Effectiveness of Acetylsalicylic Acid in Primary Prevention of Cardiovascular Diseases and Mortality in Patients at High Risk, a Retrospective Cohort Study—The JOANA Study

**DOI:** 10.3390/jcm14165710

**Published:** 2025-08-12

**Authors:** Lia Alves-Cabratosa, Carles López, Maria Garcia-Gil, Èric Tornabell-Noguera, Marc Comas-Cufí, Jordi Blanch, Ruth Martí-Lluch, Anna Ponjoan, Gina Domínguez-Armengol, Lluís Zacarías-Pons, Francesc Ribas-Aulinas, Elisabet Balló, Rafel Ramos

**Affiliations:** 1Vascular Health Research Group (ISV-Girona), Institut Universitari d’Investigació en Atenció Primària Jordi Gol (IDIAP Jordi Gol), 17001 Girona, Catalonia, Spain; lalves@idiapjgol.org (L.A.-C.); gdominguez@idiapjgol.org (G.D.-A.);; 2Network for Research on Chronicity, Primary Care, and Prevention and Health Promotion (RICAPPS), 17001 Girona, Catalonia, Spain; 3Serveis d’Atenció Primària, Institut Català de Salut (ICS), 17001 Girona, Catalonia, Spain; clarpi.girona.ics@gencat.cat (C.L.);; 4Computer Science, Applied Mathematics and Statistics Department, Universitat de Girona, 17003 Girona, Catalonia, Spain; 5Biomedical Research Institute Girona (IdIBGi), 17190 Girona, Catalonia, Spain; 6Departament de Ciències Mèdiques, Universitat de Girona, 17071 Girona, Catalonia, Spain

**Keywords:** aspirin, primary prevention, age groups, real-world data

## Abstract

**Background/Objectives:** Although differences seem to exist by age in primary cardiovascular prevention with acetylsalicylic acid (ASA), direct comparisons are lacking, as are studies with real-world data. We sought to examine the effectiveness of ASA in reducing cardiovascular diseases and overall mortality in patients at high risk by age subgroups. **Methods:** We designed a retrospective cohort study using the database of the Catalan primary care system (SIDIAP), Spain, for the period 2006–2020. Included participants were high-cardiovascular-risk individuals without previous vascular disease. We considered people aged 40 to 59 and ≥60 years of age. We assessed the incidences of atherosclerotic cardiovascular disease (ASCVD), all-cause mortality, and ASA adverse effects using Cox proportional hazards modelling, adjusted by the propensity score of ASA treatment. **Results:** During the study period, 7576 and 30,282 people were aged 40 to 59 and ≥60 years, respectively. The median follow-up was 11.21 (10.71–11.54) years (40 to 59 year-olds) and 11.09 (10.55–11.54) years (≥60 year-olds). The hazard ratio of ASA use for ASCVD in the group aged 40–59 years was 0.64 (0.41–0.99). The number needed to treat in this group was 40 persons and the number that needed to harm for gastrointestinal bleeding (the only adverse effect with significant hazard ratio) was 75 individuals. **Conclusions:** This direct comparison of real-world age groups at high cardiovascular risk showed no benefit but increased risk in the older population (≥60 years). In the younger subgroup, our observations would support primary prevention with ASA with a consideration of the individual optimal risk–benefit.

## 1. Introduction

Cardiovascular diseases (CVDs) are the leading cause of death globally [1]. In Europe, the incidence of ischemic heart disease was 5.8 million cases in 2019 [2,3], and in 2021, ischaemic heart disease (IHD) accounted for 45% deaths in women, and 39% in women and men, respectively [3]. In Spain, CVDs were reported as the cause of death in 69,483 persons in 2023 [4]. The impact of these diseases is huge and includes both direct effects on cardiovascular health and implications on the quality of life [5].

The Guidelines on cardiovascular prevention consider it essential to target populations at high risk, because interventions directed to reduce the level of the risk factors are more efficient in this population than in the general or in the low-risk populations [6]. Treatment with low doses of acetylsalicylic acid (ASA) is one of the measures for cardiovascular prevention. It is well established in secondary prevention [6], but its use in primary prevention is more controversial. While current evidence does not suggest differences by sex [7], the Guidelines do present a variety of considerations by age subgroups and cardiovascular risk [6,8,9].

The European Society of Cardiology (ESC) [6] and the National Institute for Health and Care Excellence (NICE) [8] Guidelines advocate against the routine use of aspirin for patients without established atherosclerotic cardiovascular disease (ASCVD) and do not specify recommendations by age. However, the first states that the benefit might outweigh the risks in some patients at high or very high cardiovascular risk. The American College of Cardiology (ACC)/American Heart Association (AHA) [9] considers the use of low-dose aspirin as primary prevention in the population at high CVD risk, but differs from the latest United States Preventive Services Task Force (USPSTF) [7] recommendations on the age threshold to discourage such use due to the increase in the risk of bleeding risk and the subsequent loss of net benefit.

The latest recommendations are based on the inclusion of updated randomised clinical trials—in higher-risk populations—within more recent meta-analysis [10,11,12] at a time when the use of statins for the primary prevention of CVD is preferential. However, very few studies have addressed the effectiveness of ASA using real-world data [13], which complement the evidence from the controlled conditions of the trials with that from daily clinical practice, reflecting the variety of its users, including their comorbidities, behaviour, and priorities.

The USPSTF emphasises that the decision to initiate low-dose ASA in primary cardiovascular prevention should be an individual one, taking both ischaemic and bleeding risks into consideration, a frequent statement in most of the studies and guidelines [7,14]. They also set a threshold at 60 years or older to recommend against initiating ASA treatment [15]. However, the level of evidence for those aged 40 to 59 years is less robust, classified as level C. Moreover, no studies have directly compared the effectiveness of ASA initiation for primary cardiovascular prevention in high-risk populations considering the mentioned age ranges. Accordingly, we sought to assess the effectiveness and safety of ASA in reducing cardiovascular diseases in high-risk patients by age group, using real-world data with several years of follow-up.

## 2. Materials and Methods

### 2.1. Data Source

The Information System for the Development of Research in Primary Care (SIDIAP) is a clinical database of anonymised longitudinal patient records of more than six million people (80% of the Catalan population and 10% of the total population of Spain) registered in 274 primary care practices that included 3414 general practitioners at the time of the study [16]. The recorded information encompasses demographic and lifestyle factors relevant to primary care settings (e.g., body mass index, smoking status, alcohol use); clinical diagnoses and outcomes (coded according to the International Classification of Diseases, 10th revision [ICD-10]); referrals and hospital discharge information (international classification of diseases, 9th revision [ICD-9]); laboratory tests; and prescribed medications (coded according to the Anatomical Therapeutic Chemical [ATC] classification) that have been dispensed by community pharmacies. The high quality of SIDIAP data has been previously evidenced and the database has been widely used to study the epidemiology of a number of health outcomes [17,18,19]. The investigations were conducted in accordance with the principles outlined in the Declaration of Helsinki (1975, revised in 2013). Ethics approval for observational research using SIDIAP data was obtained from the local ethics committee (P13/096) on 26 January 2022, ensuring that the study adheres to both national and international guidelines.

### 2.2. Study Design

We carried out a retrospective cohort study in the context of the Joint Data on Aspirine and Health Outcomes (JOANA) project. Enrolment was from July 2008 until December 2009. For ASA users, index date was the date of the first ASA invoice within enrolment period; for non-users, it was selected at random according to the distribution of the index date in users. Follow-up lasted until the occurrence of an outcome, transfer from SIDIAP, or the end of the study period, in December 2020.

### 2.3. Study Population

The study included people aged ≥40 years at high cardiovascular risk without previous vascular disease. Candidates had at least one visit recorded in the electronic medical records during the 18 months before the index date. We considered that individuals were at high cardiovascular risk if they had a Framingham-*Registre Gironí del Cor* (heart registry from Girona, REGICOR) [20] risk score ≥10%. We excluded individuals with previous history of CVD, defined as any of the following conditions: symptomatic peripheral arterial disease, ischemic stroke, heart failure, and coronary heart disease (CHD)—which included non-fatal angina, non-fatal myocardial infarction, or cardiac revascularisation. We also excluded participants who were taking medications to treat cardiac diseases (Anatomical Therapeutic Chemical code C01), and, to avoid frailty bias, individuals with cancer, dementia, or paralysis; and those in dialysis, institutionalised, or who had received an organ transplant.

### 2.4. ASA Exposure, Outcomes, and Covariates

To prevent survivor bias and covariate measurement bias, a “new-users design” was selected over “all ASA users” [21]. New user was any person who received an ASA treatment for the first time ever, or a person who initiated ASA treatment with no such pharmacy invoicing recorded during the previous 18 months. At least two ASA invoices during the enrolment period were required. Exposure to ASA was calculated according to the medication possession ratio (MPR) and analysis was considered for the population with a one-year MPR ≥80%. Dose was defined as the number of dispensed packages.

The onset of CVD during follow-up was identified from relevant SIDIAP codes in both primary care and hospital discharge records. Primary outcomes were total mortality and ASCVD, a composite of CHD (fatal and non-fatal angina, fatal and non-fatal myocardial infarction or cardiac revascularisation) and stroke (fatal and nonfatal ischemic stroke). We also considered CHD and ischemic stroke separately, as secondary outcomes. Finally, we assessed the following adverse effects of treatment with ASA: gastric ulcer, gastrointestinal bleeding, and haemorrhagic stroke.

We explored the variables associated with ASA prescription to determine candidate variables for the propensity score (PS) of ASA treatment. The PS was used to equalise the baseline characteristics of ASA non-users and new users in such a way that these two groups would only differ regarding ASA initiation. Details on the PS building are explained in Section A.1.1 of the Section A.1. Methods.

### 2.5. Statistical Analyses

All analyses of this high-risk population considered two age subgroups: people aged 40 to 59 years and ≥60 years. Categorical variables were presented as numbers (percentages) and continuous variables as means (standard deviation or their 95% confidence intervals), or median (first and third quartiles), as appropriate.

We used 20 multiple imputations by chained equations [22] to replace the missing baseline values of total cholesterol, low-density lipoprotein cholesterol, high-density lipoprotein cholesterol, triglycerides, glucose, systolic, and diastolic blood pressure, and body mass index (BMI). The validation of the imputation process is explained in Section A.1.2 of the Section A.1. Methods.

Because of the non-random treatment allocation, a logistic model based on potential confounding covariates was used to calculate the PS of ASA therapy. The PS was separately calculated within each age group. Standardised differences before and after adjusting for PS were calculated and variables with adjusted differences <0.10 were considered well-balanced [23]; further details are explained in Section A.1.2 of the Section A.1. Methods.

The hazard ratios (HRs) of exposure to ASA were calculated for the outcome events using Cox proportional hazard regression models adjusted by the PS of ASA initiation. Additional adjustments were performed to prevent residual confounding: variables that remained imbalanced after PS adjustment (i.e., with a standardised difference between non-users and new users ≥0.10) were also included in the models. This two-step approach ensures that confounding is adequately addressed, preventing bias in the hazard ratio estimates [24]. The proportionality of hazards assumption was tested. Participants were censored at the date of transfer from SIDIAP or end of the study period.

We calculated the incidences and 95% CI of each outcome using Poisson models. The 5-year numbers needed to treat (NNT) and to harm (NNH) for one additional patient to survive without reaching an endpoint were also calculated for the outcomes with significant HRs. The reference individual for the NNT-NNH calculation was a woman with the mean values of BMI, total cholesterol, glucose, triglycerides, creatinine levels, and height; no smoking habit, dyslipidaemia, benign neoplasms, or arthritis, and not treated with psychoanaleptics or anti-inflammatory and antirheumatic products. We analysed the data using a simulated “intention-to-treat” scenario where subsequent treatment changes in the exposed and unexposed patients did not modify the category of exposure or study ending time. All statistical analyses were carried out using R-software 4.5.0 [25,26].

## 3. Results

During the study period, 37,858 individuals met the inclusion criteria, 7576 of them were aged 40 to 59 years, and 30,282 of them were aged ≥60 years. The study flowchart is detailed in Figure 1.

The median (interquartile range—IQR) follow-up was 11.21 (10.71–11.54) years for participants aged 40 to 59 years and 11.09 (10.55–11.54) years for those aged ≥60 years. A total of 315 (4.16%) participants in the younger group and 314 (1.04%) in the older group were lost to follow-up (transferred out of the SIDIAP reference area). Participants’ mean (standard deviation—SD) ages in the groups <60 and ≥60 years were 55.7 (3.5) and 68.2 (4.3) years, respectively. Women represented approximately 23.4% of participants <60 years and 16.3% of those ≥60 years of age. The description of the variables with missing values is presented in the Appendix A Table A1.

The health status was slightly better in the older population, who had slightly lower blood pressure levels and BMI, as well as lower percentages of people with smoking habit and high alcohol consumption. They also had a better lipid profile. Finally, treatment with statins (or other lipid-lowering drugs) concomitant with ASA was more frequent in the younger subgroup. The baseline characteristics for both ASA new-users and non-users are compared in Table 1. The baseline characteristics of the complete-cases subset are compared in the Appendix A Table A2.

In people over 60 years of age, there were no protective associations between ASA treatment initiation and the considered cardiovascular outcomes or mortality. The HRs for the adverse effects were significant for gastrointestinal bleeding (1.56 (1.04–2.32)) and gastric ulcer (1.69 (1.02–2.81)). The HR of ASA use in participants aged 40–59 years for ASCVD was 0.64 (0.41–0.99) (Figure 2).

In this group (aged 40–59 years), no significant associations with the secondary outcomes (CHD and ischaemic stroke) or mortality were observed. Regarding the adverse effects, gastrointestinal bleeding presented significant HRs, which were 2.77 (1.34–5.72). The 5-year NNTs for ASCVD in these 40-to-59-year-olds was 40 individuals, lower than the NNH for gastrointestinal bleeding, which was 75 people. Table 2 shows the unadjusted incidences for mortality and incident cardiovascular events by age groups. The incidences and HRs of the complete-cases subset are presented in the Section A.2, Table A3.

## 4. Discussion

Our analysis of real-world data compared age groups at high cardiovascular risk but no previous cardiovascular events (primary prevention) and found no protective association between ASA use and cardiovascular events in individuals aged at least 60 years old. Moreover, ASA initiation was associated with an increased risk of gastrointestinal bleeding and gastric ulcer, findings that clearly argue against endorsing such a preventive treatment in this age group. In the population aged 40 to 59 years, we observed a protective relative association of ASA initiation with the occurrence of ASCVD and also an increased risk of gastrointestinal bleeding. The examination of the absolute differences favoured treatment: the NNT for ASCV and the NNH for gastrointestinal bleeding were 40 and 75 participants, respectively. However, the decision to treat should be on an individual basis and require a cautious balance of benefits and potential harms.

Previous meta-analysis of clinical trials suggested that the major benefits of aspirin treatment may be offset by its harmful effects when added to other primary prevention medications such as statins [27]. This was the basis for Barnett et al. (2010) to recommend against the initiation of ASA treatment in the primary prevention of cardiovascular disease [28]. Overall, the antithrombotic trialists (ATTs) collaboration found a 12% reduction in any serious vascular event, which translated into a small absolute reduction, of 0.06% per year [27]. However, most of the population in the meta-analysis were not receiving statin treatment at the time of the publication, which would currently be a preferred therapy to ASA in cardiovascular primary prevention. The authors consider that the absolute benefit of treatment could be halved by concomitant statin therapy, while the main bleeding hazards—reported as 0.03% per year in the meta-analysis—would persist, thereby balancing the absolute benefits of aspirin treatment against the risks.

The ATT collaboration also highlighted the possibility of obtaining a net benefit with aspirin treatment in certain subgroups of individuals [29]. In this regard, they analysed the results by age and by risk groups (separately) in the primary prevention population. By age, the group of <65-year-olds had a significant protective effect of aspirin (rate ratio [95%CI]: 0.87 [0.78–0.98]), very similar to that in the group aged ≥65 years, which was non-significant (rate ratio: 0.88 [0.77–1.01]). By risk, the authors questioned the reliability of their non-significant findings because of the small number of participants in the highest risk category. In any case, they emphasised the importance of further analysis considering subgroups by age and cardiovascular risk. In line with this, Kim et al. (2021) also advocated for evaluating all the absolute benefits and risks—not only age [27]. Their perspective incorporated more recent trial data that aimed to assess individuals at high cardiovascular risk. They concluded that benefits would likely outweigh risks in the population with an absolute cardiovascular risk >10%.

Another analysis, which encompassed the ATT collaboration and the more recent trials mentioned above, was conducted by Marquis-Gravel et al. (2019). They found no significant relative benefit from aspirin (RR: 0.98 [0.93–1.02]) and a significant harmful effect (RR: 1.47 [1.31–1.65]) [30]. The populations from the trials varied. Particularly, the Aspirin in Reducing Events in the Elderly (ASPREE) trial [12] included people aged ≥65 and 70 years, depending on their ethnicity. They found no significant benefit from aspirin in reducing the incidence of cardiovascular disease and evidenced a significant increase in all-cause mortality, amounting to 1 additional death per 1000 persons per year. Their conclusions were similar when considering compliance-adjusted effects [31]. Thus, their recommendation for this older population was against aspirin for primary prevention. More recently, Chowdhury et al. (2025) presented an analysis stratified by cardiovascular risk in older adults and reached similar conclusions: they could not identify cardiovascular risk subgroups of older adults who might benefit from aspirin treatment [32]. Although this remains an ongoing debate [33], our results for the older population at high risk are consistent with these findings and with the results of the ASPREE trial [12].

The most recent statement from the US Preventive Services Task Force includes considerations by age and risk [7]. The authors concluded that aspirin would be associated with a decreased risk of first cardiovascular event (myocardial infarction and stroke) but not with reduced mortality. Regarding adverse effects, they highlighted a considerable relative increase in the risk of bleeding, particularly of major gastrointestinal bleeding and intracranial bleeds. They also noted no differences in the relative effect of aspirin by age. However, when estimating the magnitude of the net gain, they reported differences by age and risk, in terms of quality of life and life-years. Considering these absolute measures, they stated that people with a 10-year CVD risk ≥10% who are aged 40 to 59 years would generally benefit from initiating aspirin treatment. In people aged 60 years or over, and especially in people aged ≥70 years, the harms of such treatment (expressed in loss of quality-adjusted life-years and life-years) would outweigh its benefits. Overall, our results are also consistent with these recommendations and provide a direct comparison that contributes to supporting them.

Previous studies have assessed subgroups of age and risk separately but, to the best of our knowledge, this is the first analysis that examines a population at high risk and directly compares age subgroups. In the population ≥60 years of age, the lack of protective association of ASA use and the increased risk of adverse events argues against the initiation of this treatment. In the younger population, the absolute differences between NNT and NNH support primary prevention with ASA in this group. However, an assessment of the risk–benefit balance is essential, and thus, clinical judgement remains central to individualised decision making in this age group. Our results provide real-world evidence that aligns with the findings from several of the randomised controlled trials mentioned above and the recommendations from the US Preventive Services Task Force.

### Study Characteristics That Merit Consideration

Electronic medical records offer a complementary perspective for addressing relevant questions, particularly those related to the effectiveness of medical treatments, because they reflect actual clinical practice (and can be carried out at a reasonable cost). Data must be collected in well-designed, quality-assured databases and analysed using appropriate methods [34]. The use of data from electronic databases is a noteworthy aspect of modern epidemiology in this 21st century [35], because it can replicate the effects observed in randomised controlled trials [36]. However, this type of observational studies could tend to overestimate the effect size of certain interventions [37] if several key points are not well addressed. First, poor data quality could generate misclassification. In this study, not only was the representativeness of the data assessed—based on the geographical, age, and sex distributions of the population of Catalonia—but the data were also validated for cardiovascular risk factors and CVD [17]. Moreover, the validity of the recorded ASA exposure was confirmed using official invoicing records from community pharmacies. Nevertheless, we cannot exclude some degree of outcome underreporting. This could lead to a non-differential misclassification, reducing statistical power, and potentially biasing the results, albeit towards the null hypothesis.

Second, confounding by indication can produce bias observational studies. However, we used a new user design to estimate confounding factors. We then adjusted for the PS, and further adjusted for variables that remained unbalanced after including the PS in the models. Third, the presence of missing data can influence the results. We applied multiple imputation rather than excluding individuals with missing values, to avoid a selection bias that may occur if the population with missing data differs from the complete cases. Fourth, immortal bias can occur when the determination of the treatment status of a given individual involves a delay during which, by design, death or the study outcome cannot be assessed. To address this, we applied prescription time-distribution matching in addition to PS adjustment [38]: the distribution of the non-users’ index date was matched to the timing of first prescription in users. This approach prevents the imbalance of the prescription time distribution between the two groups. Finally, while the lower population size within the high-risk ASA-treated subgroup may have reduced the statistical power of the analysis, it reflects the specific population group targeted for this intervention.

In conclusion, a direct comparison of age subgroups at high cardiovascular risk provided evidence from the real-world settings against the initiation of ASA treatment in the population aged 60 years or older. Not only did this treatment show no benefit in reducing cardiovascular events among older participants, but it was also associated with a significant increase in adverse effects. The evidence for the younger population (40 to 59 years of age) yielded an NNT for ASCVD slightly lower than the NNH for gastrointestinal bleeding. All in all, our observations would support the consideration of primary prevention with ASA in the younger group, although the optimal risk–benefit should be considered on an individual basis.

## Figures and Tables

**Figure 1 jcm-14-05710-f001:**
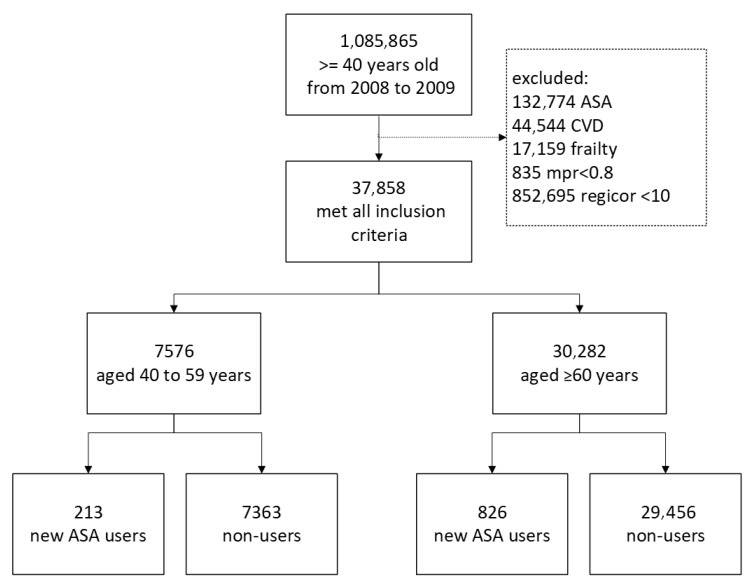
Study flowchart. ASA, acetylsalicylic acid; CVD, cardiovascular disease; mpr, medication possession ratio.

**Figure 2 jcm-14-05710-f002:**
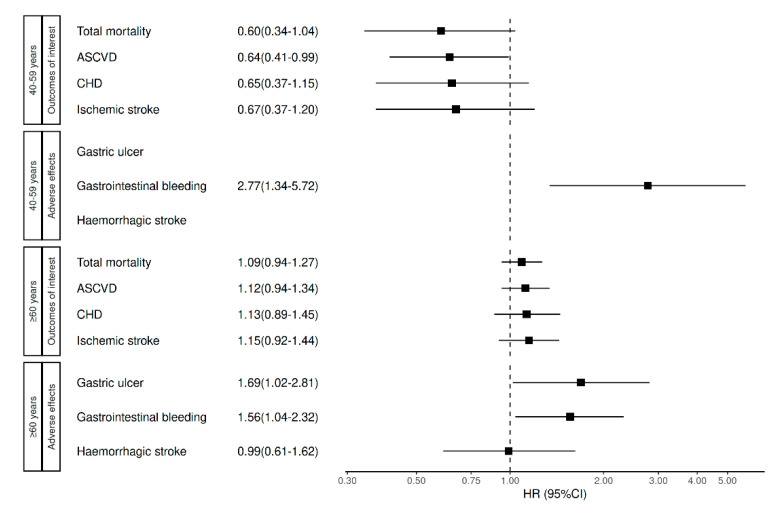
Adjusted hazard ratios of ASA for mortality and incident cardiovascular events by age groups. ASCVD, atherosclerotic cardiovascular disease; and CHD, coronary heart disease.

**Table 1 jcm-14-05710-t001:** Baseline characteristics of the study population by age groups.

	40–59 Years				≥60 Years			
	ASA Non-Users *n* = 7363	ASA Users *n* = 213	Standardised Difference	Adjusted Standardised Difference	ASA Non-Users *n* = 29,456	ASA Users *n* = 826	Standardised Difference	Adjusted Standardised Difference
Age	55.72 (3.5)	55.34 (3.9)	0.1081	0.1010	68.18 (4.3)	67.69 (4.2)	0.1136	0.1355
Men	76.8%	71.96%	0.1076	0.1892	89.58%	82.97%	0.1743	0.0069
Systolic blood pressure	145.67 (14.5)	147.07 (15.7)	−0.0966	0.1018	142.99 (15)	146.06 (17)	−0.2043	0.0613
Diastolic blood pressure	86.01 (9.9)	84.7 (10.6)	0.1316	0.0882	81.64 (9.7)	80.77 (9.8)	0.0890	0.0747
BMI	30.42 (4.6)	30.83 (5)	−0.0872	0.0089	29.28 (4.4)	29.87 (4.5)	−0.1355	0.0775
Vascular risk factors								
DM	45%	85.89%	−0.9840	0.0000	32.29%	72.99%	−0.8097	0.0000
Hypertension	41.53%	55.91%	−0.2855	0.0228	44.59%	68.43%	−0.4914	0.0002
Smoking	58.97%	52.67%	0.1259	0.0880	42.72%	36.05%	0.1383	0.0132
High alcohol consumption	11.32%	11.74%	−0.0132	0.0151	7.47%	8.24%	−0.0282	0.0267
Other comorbidities								
Arthritis	0.56%	0.02%	0.1851	0.0988	0.6%	0.61%	−0.0017	0.0365
Asthma	2.04%	0.52%	0.1895	0.0786	2%	1.77%	0.0174	0.0146
Hypothyroidism	2.08%	3.59%	-0.0806	0.0133	1.96%	2.25%	−0.0195	0.0854
Other medications								
Statins	18.79%	53.56%	−0.6310	0.0000	20.6%	51.3%	−0.5693	0.0000
Other lipid-lowering drugs	24.83%	66.81%	−0.7824	0.0000	24.03%	57.74%	−0.6275	0.0000
Diuretics	17.63%	32.96%	−0.3185	0.0001	22.71%	44.86%	−0.4283	0.0001
Beta-blockers	6.49%	14.37%	−0.2189	0.0041	6.98%	12.8%	−0.1719	0.0126
Calcium channel blockers	6.37%	12.35%	−0.1790	0.0000	8.16%	20.41%	−0.2924	0.0002
ACEI	27.84%	57.36%	−0.5611	0.0000	32.3%	64.12%	−0.6179	0.0000
Anti-diabetics	27.18%	74.24%	−0.9088	0.0000	20.82%	65.54%	−0.8064	0.0000
Anti-inflammatory drugs	22.59%	27.89%	−0.1182	0.1311	24.52%	28.93%	−0.0972	0.1486
Laboratory tests								
Total cholesterol	240.36 (40.8)	236.39 (45.1)	0.0973	0.2283	223.98 (38.1)	219.94 (38.3)	0.1060	0.1388
HDL cholesterol	39.35 (8.1)	39.55 (8.4)	−0.0249	0.0143	44.24 (10)	43.6 (9.2)	0.0651	0.0753
Triglycerides	242.17 (162.3)	277.3 (198.7)	−0.2150	0.0646	175.23 (103.6)	188.67 (111.9)	−0.1294	0.1033
Glucose	129.52 (51.5)	183.83 (71.6)	−1.0417	0.2948	116.61 (38.7)	152.95 (57.2)	−0.9231	0.3315
Glomerular filtration rate	86.12 (16)	85.86 (16.2)	0.0166	0.0259	78.43 (14.8)	76.85 (16.9)	0.1066	0.0212

Continuous variables are presented as mean (standard deviation), and categorical variables as percentages. ASA, acetylsalicylic acid; ACEI, angiotensin-converting enzyme inhibitors; BMI, body mass index; DM, diabetes mellitus; HDL, high-density lipoprotein.

**Table 2 jcm-14-05710-t002:** Unadjusted incidences of ASA for mortality and incident cardiovascular events by age groups.

40–59 Years	ASA Non-Users *n* = 7363		ASA New Users *n* = 213	
	Number of Events	Incidence Rate/1000 Person-Years (95% CI)	Number of Events	Incidence Rate/1000 Person-Years (95% CI)
Outcomes of interest				
Total mortality	3559	8.88 (8.59–9.18)	31	6.58 (4.63–9.35)
ASCVD	4309	11.38 (11.04–11.72)	58	13.2 (10.2–17.07)
CHD	2439	6.29 (6.04–6.54)	33	7.3 (5.19–10.27)
Ischemic stroke	2094	5.36 (5.13–5.59)	31	6.8 (4.78–9.67)
Adverse effects				
Gastric ulcer	437	1.11 (1.01–1.21)	5	1.07 (0.45–2.57)
Gastrointestinal bleeding	583	1.46 (1.35–1.59)	16	3.46 (2.12–5.65)
Haemorrhagic stroke	304	0.76 (0.68–0.85)	1	0.21 (0.03–1.51)
**≥60 Years**	**ASA Non-Users** ** *n* ** **= 29,456**		**ASA New Users** ** *n* ** **= 826**	
	**Number of Events**	**Incidence Rate/1000 Person-Years (95% CI)**	**Number of Events**	**Incidence Rate/1000 Person-Years (95% CI)**
Outcomes of interest				
Total mortality	19,084	22.32 (22.01–22.64)	340	26.42 (23.76–29.39)
ASCVD	9284	11.46 (11.23–11.7)	244	21.02 (18.54–23.83)
CHD	4490	5.40 (5.24–5.56)	118	9.63 (8.04–11.53)
Ischemic stroke	5310	6.39 (6.22–6.56)	146	12.02 (10.22–14.14)
Adverse effects				
Gastric ulcer	892	1.06 (0.99–1.13)	22	1.74 (1.15–2.64)
Gastrointestinal bleeding	1375	1.62 (1.53–1.71)	41	3.22 (2.37–4.37)
Haemorrhagic stroke	875	1.03 (0.96–1.1)	20	1.56 (1.01–2.42)

ASA, acetylsalicylic acid; ASCVD, atherosclerotic cardiovascular disease; CHD, coronary heart disease; CI, confidence interval. Incidences are unadjusted.

## Data Availability

The datasets generated and analysed for this study are not publicly available due to legal and ethical restrictions related to data privacy but are available from the corresponding author upon reasonable request.

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
