# Peer review of "Long-Term Effectiveness of Acetylsalicylic Acid in Primary Prevention of Cardiovascular Diseases and Mortality in Patients at High Risk, a Retrospective Cohort Study—The JOANA Study"

_jcm, 2025, doi:10.3390/jcm14165710_

Round 1
Reviewer 1 Report
Comments and Suggestions for Authors
This database study demonstrates the importance of applying robust statistical techniques to reduce confounders and limit exclusion of incomplete data. While the low population size within the high risk ASA-treated subgroup raises concern for statistical power and type 2 errors, it reflects the specific population group targeted for this intervention.
Congratulations to your work, this study adds evidence to current practices, and would be an excellent addition to primary care recommendations of ASA use in younger high risk patients.
Author Response
Thank you very much for taking the time to review this manuscript. Please find the detailed responses in the attached file and the corresponding revisions/corrections highlighted/in track changes in the re-submitted files.

Reviewer 2 Report
Comments and Suggestions for Authors
I reviewed with interest the manuscript by Lia Alves-Cabratosa et al. "Long term effectiveness of ASA in primary prevention of cardiovascular diseases and mortality in patients at high risk, a retrospective cohort study. The JOANA Study". In this article, the authors once again returned to the issue of primary prevention of cardiovascular diseases by prescribing aspirin. Although many meta-analyses have already shown the inappropriateness of prescribing aspirin for this purpose, which is enshrined in most international recommendations, the authors decided to study the effect of age on the effectiveness of aspirin using real-life clinical practice data. It should be noted that the authors carefully approached the study design, they used an extensive regional database, as well as well-thought-out statistical procedures. This approach yielded interesting results that can be used in further research and the development of practical recommendations.
Comments and questions when reviewing the article:
1. It is advisable to consider in the Introduction a study with a similar design, which conducts a retrospective analysis of the VITAL Cohort (ref. 1, see below).
2. A significant limitation of the study is its retrospective nature. Therefore, it is advisable to consider in the article a study with prospective prescription of aspirin for primary prevention (ref. 2, see below)
3. The aspirin group over 60 years of age differs significantly from the no-aspirin group in a number of indicators (comorbidity, medication intake), so these factors can also significantly affect the prognosis of patients. How do the authors take into account the influence of these factors in the aspirin/non-aspirin groups?
4. In the Discussion section, the authors state: "This was the basis for Barnett et al. to recommended against the initiation of ASA treatment in primary prevention of cardiovascular disease [27]" - line 227-229). However, the cited source (27) is not a publication of Barnett et al. A reference to this article should be added.
References:
1. Caldeira D, Alves M, Gonçalves N, Costa J, Ferreira JJ, Pinto FJ. Association of Aspirin Use in Primary Prevention and Cardiovascular Events: A Retrospective Analysis of the VITAL Cohort. J Pers Med. 2025 Feb 26;15(3):89. doi: 10.3390/jpm15030089.
2. Smith CL, Kasza J, Woods RL, Lockery JE, Kirpach B, Reid CM, Storey E, Nelson MR, Shah RC, Orchard SG, Ernst ME, Tonkin AM, Murray AM, McNeil JJ, Wolfe R. Compliance-Adjusted Estimates of Aspirin Effects Among Older Persons in the ASPREE Randomized Trial. Am J Epidemiol. 2023 Nov 10;192(12):2063-2074. doi: 10.1093/aje/kwad168.
Author Response
Thank you very much for taking the time to review this manuscript. Please find the detailed responses below and the corresponding revisions/corrections highlighted/in track changes in the re-submitted files.

Reviewer 3 Report
Comments and Suggestions for Authors
In this manuscript titled “Long term effectiveness of ASA in primary prevention of cardiovascular diseases and mortality in patients at high risk, a retrospective cohort study. The JOANA Study”, the authors, using a retrospective cohort study, aimed to examine the effectiveness of ASA in reducing cardiovascular diseases and overall mortality in patients at high risk classified based on age.
This study was well-executed and its findings were thoroughly and thoughtfully analyzed.
Below are a few minor comments:
Title: Please use the expanded form for ASA in the title.
Explain “The JOANA study”
Fig. 1. Please provide list of abbreviations below the flowchart like the list included below the tables (e.g., ASA, CVD, mpr, regicor)
Table 1. Please format it so that the numbers in parentheses are in the second line. Also, mention what is included in the parentheses (standard deviation?)
Table 2: Suggestion: Hazard ratio as a graphical representation, instead of numbers in table format, will improve readability.
Line 229: Please expand abbreviations at first use (e.g., ATT in line 229)
Recent publications relevant for discussion:
Chowdhury EK, Ernst ME, Nelson MR, Beilin LJ, Neumann JT, Tonkin A, Woods RL, Stocks N, Lacaze P, Orchard SG, Zhou Z, Kirpach B, Murray AM, Shah RC, Abhayaratna W, Ryan J, McNeil JJ, Wolfe R, Reid CM. Stratification by CVD risk equations does not inform the use of aspirin for primary prevention in older adults. Eur J Prev Cardiol. 2025 Jun 6:zwaf329. doi: 10.1093/eurjpc/zwaf329. Epub ahead of print. PMID: 40478248
Wittes J, DeMets DL, Kim K, Maki DG, Pfeffer MA, Gaziano JM, Kitsantas P, Hennekens CH, Wood SK. Aspirin in primary prevention: Undue reliance on an uninformative trial led to misinformed clinical guidelines. Clin Trials. 2025 Apr 1:17407745251324866. doi: 10.1177/17407745251324866. Epub ahead of print. PMID: 40165541.
Comments on the Quality of English LanguageA few corrections needed in sentence formation.
Author Response

(The authors gave the same response as above.)
